# Improving PAC Exploration
# Using the Median of Means

**Jason Pazis**
Laboratory for Information and Decision Systems
Massachusetts Institute of Technology
Cambridge, MA 02139, USA
`jpazis@mit.edu`

**Ronald Parr**
Department of Computer Science
Duke University
Durham, NC 27708
`parr@cs.duke.edu`

**Jonathan P. How**
Aerospace Controls Laboratory
Department of Aeronautics and Astronautics
Massachusetts Institute of Technology
Cambridge, MA 02139, USA
`jhow@mit.edu`

## Abstract

We present the first application of the median of means in a PAC exploration algorithm for MDPs. Using the median of means allows us to significantly reduce the dependence of our bounds on the range of values that the value function can take, while introducing a dependence on the (potentially much smaller) variance of the Bellman operator. Additionally, our algorithm is the first algorithm with PAC bounds that can be applied to MDPs with unbounded rewards.

## 1    Introduction

As the reinforcement learning community has shifted its focus from heuristic methods to methods that have performance guarantees, PAC exploration algorithms have received significant attention. Thus far, even the best published PAC exploration bounds are too pessimistic to be useful in practical applications. Even worse, lower bound results [14, 7] indicate that there is little room for improvement.

While these lower bounds prove that there exist pathological examples for which PAC exploration can be prohibitively expensive, they leave the door open for the existence of "well-behaved" classes of problems in which exploration can be performed at a significantly lower cost. The challenge of course is to identify classes of problems that are general enough to include problems of real-world interest, while at the same time restricted enough to have a meaningfully lower cost of exploration than pathological instances.

The approach presented in this paper exploits the fact that while the square of the maximum value that the value function can take ($Q_{\max}^2$) is typically quite large, the variance of the Bellman operator is rather small in many domains of practical interest. For example, this is true in many control tasks: It is not very often that an action takes the system to the best possible state with $50\%$ probability and to the worst possible state with $50\%$ probability.

Most PAC exploration algorithms take an average over samples. By contrast, the algorithm presented in this paper splits samples into sets, takes the average over each set, and returns the median of the averages. This seemingly simple trick (known as the median trick [1]), allows us to derive sample complexity bounds that depend on the variance of the Bellman operator rather than $Q_{\max}^2$. Addi-

tionally, our algorithm (Median-PAC) is the first reinforcement learning algorithm with theoretical guarantees that allows for unbounded rewards.[1]

Not only does Median-PAC offer significant sample complexity savings in the case when the variance of the Bellman operator is low, but even in the worst case (the variance of the Bellman operator is bounded above by $\frac{Q_{\max}^2}{4}$) our bounds match the best, published PAC bounds. Note that Median-PAC does not require the variance of the Bellman operator to be known in advance. Our bounds show that there is an inverse relationship between the (possibly unknown) variance of the Bellman operator and Median-PAC's performance. This is to the best of our knowledge not only the first application of the median of means in PAC exploration, but also the first application of the median of means in reinforcement learning in general.

Contrary to recent work which has exploited variance in Markov decision processes to improve PAC bounds [7, 3], Median-PAC makes no assumptions about the number of possible next-states from every state-action (it does not even require the number of possible next states to be finite), and as a result it is easily extensible to the continuous state, concurrent MDP, and delayed update settings [12].

## 2   Background, notation, and definitions

In the following, important symbols and terms will appear in **bold** when first introduced. Let $\mathcal{X}$ be the domain of $x$. Throughout this paper, $\forall x$ will serve as a shorthand for $\forall x \in \mathcal{X}$. In the following $s, \bar{s}, \tilde{s}, s'$ are used to denote various states, and $a, \bar{a}, \tilde{a}, a'$ are used to denote actions.

A *Markov Decision Process* (MDP) [13] is a 5-tuple $(\mathcal{S}, \mathcal{A}, P, R, \gamma)$, where $\mathcal{S}$ is the state space of the process, $\mathbf{A}$ is the action space[2], $\mathbf{P}$ is a Markovian transition model $\big(p(s'|s, a)$ denotes the probability of a transition to state $s'$ when taking action $a$ in state $s\big)$, $\mathbf{R}$ is a reward function $\big(R(s, a, s')$ is the reward for taking action $a$ in state $s$ and transitioning to state $s'\big)$, and $\boldsymbol{\gamma} \in [0, 1)$ is a discount factor for future rewards. A *deterministic policy* $\boldsymbol{\pi}$ is a mapping $\pi : \mathcal{S} \mapsto \mathcal{A}$ from states to actions; $\pi(s)$ denotes the action choice in state $s$. The value $\boldsymbol{V^\pi(s)}$ of state $s$ under policy $\pi$ is defined as the expected, accumulated, discounted reward when the process begins in state $s$ and all decisions are made according to policy $\pi$. There exists an optimal policy $\boldsymbol{\pi^*}$ for choosing actions which yields the optimal value function $V^*(s)$, defined recursively via the Bellman optimality equation $\boldsymbol{V^*(s)} = \max_a \left\{ \sum_{s'} p(s'|s, a) \left( R(s, a, s') + \gamma V^*(s') \right) \right\}$. Similarly, the value $\boldsymbol{Q^\pi(s, a)}$ of a state-action $(s, a)$ under policy $\pi$ is defined as the expected, accumulated, discounted reward when the process begins in state $s$ by taking action $a$ and all decisions thereafter are made according to policy $\pi$. The Bellman optimality equation for $Q$ becomes $\boldsymbol{Q^*(s, a)} = \sum_{s'} p(s'|s, a) \left( R(s, a, s') + \gamma \max_{a'} \{ Q^*(s', a') \} \right)$. For a fixed policy $\pi$ the Bellman operator for $Q$ is defined as $\boldsymbol{B^\pi Q(s, a)} = \sum_{s'} p(s'|s, a) \Big( R(s, a, s') + \gamma Q(s', \pi(s')) \Big)$. In reinforcement learning (RL) [15], a learner interacts with a stochastic process modeled as an MDP and typically observes the state and immediate reward at every step; however, the transition model $P$ and reward function $R$ are not known. The goal is to learn a near optimal policy using experience collected through interaction with the process. At each step of interaction, the learner observes the current state $s$, chooses an action $a$, and observes the reward received $r$, and resulting next state $s'$, essentially sampling the transition model and reward function of the process. Thus experience comes in the form of $(s, a, r, s')$ samples.

We assume that all value functions $Q$ live in a complete metric space.

**Definition 2.1.** $\boldsymbol{Q_{\mathbf{max}}}$ *denotes an upper bound on the expected, accumulated, discounted reward from any state-action under any policy.*

We require that $Q_{\min}$, the minimum expected, accumulated, discounted reward from any state-action under any policy is bounded, and in order to simplify notation we also assume without loss of

generality that it is bounded below by 0. If $Q_{\min} < 0$, this assumption is easy to satisfy in all MDPs for which $Q_{\min}$ is bounded by simply shifting the reward space by $(\gamma - 1)Q_{\min}$.

There have been many definitions of **sample complexity** in RL. In this paper we will be using the following [12]:

**Definition 2.2.** *Let* $(s_1, s_2, s_3, \dots)$ *be the random path generated on some execution of* $\pi$, *where* $\pi$ *is an arbitrarily complex, possibly non-stationary, possibly history dependent policy (such as the policy followed by an exploration algorithm). Let* $\epsilon$ *be a positive constant,* $T$ *the (possibly infinite) set of time steps for which* $V^\pi(s_t) < V^*(s_t) - \epsilon$, *and define*[3]

$$\epsilon_e(t) = V^*(s_t) - V^\pi(s_t) - \epsilon, \ \forall \, t \in T.$$
$$\epsilon_e(t) = 0, \ \forall \, t \notin T.$$

*The Total Cost of Exploration (**TCE**) is defined as the undiscounted infinite sum* $\sum_{t=0}^{\infty} \epsilon_e(t)$.

"Number of suboptimal steps" bounds follow as a simple corollary of TCE bounds.

We will be using the following definition of efficient PAC exploration [14]:

**Definition 2.3.** *An algorithm is said to be* **efficient PAC-MDP** (*Probably Approximately Correct in Markov Decision Processes*) *if, for any* $\epsilon > 0$ *and* $0 < \delta < 1$, *its sample complexity, its per-timestep computational complexity, and its space complexity, are less than some polynomial in the relevant quantities* $(S, A, \frac{1}{\epsilon}, \frac{1}{\delta}, \frac{1}{1-\gamma})$, *with probability at least* $1 - \delta$.

## 3 The median of means

Before we present Median-PAC we will demonstrate the usefulness of the median of means with a simple example. Suppose we are given $n$ independent samples from a random variable $X$ and we want to estimate its mean. The types of guarantees that we can provide about how close that estimate will be to the expectation, will depend on what knowledge we have about the variable, and on the method we use to compute the estimate. The main question of interest in our work is how many samples are needed until our estimate is $\epsilon$-close to the expectation with probability at least $1 - \delta$.

Let the expectation of $X$ be $E[X] = \mu$ and its variance $var[X] = \sigma^2$. Cantelli's inequality tells us that: $P\left(X - \mu \geq \epsilon\right) \leq \frac{\sigma^2}{\sigma^2 + \epsilon^2}$ and $P\left(X - \mu \leq -\epsilon\right) \leq \frac{\sigma^2}{\sigma^2 + \epsilon^2}$. Let $X_i$ be a random variable describing the value of the $i$-th sample, and define $X' = \frac{X_1 + X_2 + \cdots + X_n}{n}$. We have that $E[X'] = \mu$ and $var[X'] = \frac{\sigma^2}{n}$. From Cantelli's inequality we have that $P\left(X' - \mu \geq \epsilon\right) \leq \frac{\sigma^2}{\sigma^2 + n\epsilon^2}$ and $P\left(X' - \mu \leq -\epsilon\right) \leq \frac{\sigma^2}{\sigma^2 + n\epsilon^2}$. Solving for $n$ we have that we need at most $n = \frac{(1-\delta)\sigma^2}{\delta\epsilon^2} = O\left(\frac{\sigma^2}{\delta\epsilon^2}\right)$ samples until our estimate is $\epsilon$-close to the expectation with probability at least $1 - \delta$. In RL, it is common to apply a union bound over the entire state-action space in order to prove uniformly good approximation. This means that $\delta$ has to be small enough that even when multiplied with the number of state-actions, it yields an acceptably low probability of failure. The most significant drawback of the bound above is that it grows very quickly as $\delta$ becomes smaller. Without further assumptions one can show that the bound above is tight for the average estimator.

If we know that $X$ can only take values in a bounded range $a \leq X \leq b$, Hoeffding's inequality tells us that $P\left(X' - \mu \geq \epsilon\right) \leq e^{-\frac{2n\epsilon^2}{(b-a)^2}}$ and $P\left(X' - \mu \leq -\epsilon\right) \leq e^{-\frac{2n\epsilon^2}{(b-a)^2}}$. Solving for $n$ we have that $n = \frac{(b-a)^2 \ln \frac{1}{\delta}}{2\epsilon^2}$ samples suffice to guarantee that our estimate is $\epsilon$-close to the expectation with probability at least $1 - \delta$. Hoeffding's inequality yields a much better bound with respect to $\delta$, but introduces a quadratic dependence on the range of values that the variable can take. For long planning horizons (discount factor close to 1) and/or large reward magnitudes, the range of possible $Q$-values can be very large, much larger than the variance of individual state-actions.

We can get the best of both worlds by using a more sophisticated estimator. Instead of taking the average over $n$ samples, we will split them into $k_m = \frac{n\epsilon^2}{4\sigma^2}$ sets of $\frac{4\sigma^2}{\epsilon^2}$ samples each,[4] compute the

average over each set, and then take the median of the averages. From Cantelli's inequality we have that with probability at least $\frac{4}{5}$, each one of the sets will not underestimate, or overestimate the mean $\mu$ by more than $\epsilon$. Let $f^-$ be the function that counts the number of sets that underestimate the mean by more than $\epsilon$, and $f^+$ the function that counts the number of sets that overestimate the mean by more than $\epsilon$. From McDiarmid's inequality [9] we have that $P\left(f^- \geq \frac{k_m}{2}\right) \leq e^{-\frac{2\left(\frac{3k_m}{10}\right)^2}{k_m}}$ and $P\left(f^+ \geq \frac{k_m}{2}\right) \leq e^{-\frac{2\left(\frac{3k_m}{10}\right)^2}{k_m}}$. Solving for $n$ we have that $n = \frac{\frac{200}{9}\sigma^2 \ln\left(\frac{1}{\delta}\right)}{\epsilon^2} \approx \frac{22.22\sigma^2 \ln\left(\frac{1}{\delta}\right)}{\epsilon^2}$ samples suffice to guarantee that our estimate is $\epsilon$-close to the expectation with probability at least $1 - \delta$. The median of means offers logarithmic dependence on $\frac{1}{\delta}$, independence from the range of values that the variables in question can take (even allowing for them to be infinite), and can be computed efficiently. The median of means estimator only requires a finite variance and the existence of a mean. No assumptions (including boundedness) are made on higher moments.

## 4 Median PAC exploration

---

**Algorithm 1** Median-PAC

---

1: Inputs: start state $s$, discount factor $\gamma$, max number of samples $k$, number of sets $k_m$, and acceptable error $\epsilon_a$.
2: Initialize sample sets $u_{new}(s,a) = \emptyset, u(s,a) = \emptyset \ \forall \ (s,a)$. ($|u(s,a)|$ denotes the number of samples in $u(s,a)$)
3: Set $\epsilon_b = \epsilon_a\sqrt{k}$, and initialize value function $\tilde{Q}(s,a) = Q_{\max} \ \forall \ (s,a)$.
4: **loop**
5:     Perform action $a = \arg\max_{\tilde{a}} \tilde{Q}(s,\tilde{a})$
6:     Receive reward $r$, and transition to state $s'$.
7:     **if** $|u(s,a)| < k$ **then**
8:         Add $(s,a,r,s')$ to $u_{new}(s,a)$.
9:         **if** $|u_{new}(s,a)| > |u(s,a)|$ and $|u_{new}(s,a)| = 2^i k_m$, where $i \geq 0$ is an integer **then**
10:             $u(s,a) = u_{new}(s,a)$
11:             $u_{new}(s,a) = \emptyset$
12:         **end if**
13:         **while** $max_{(s,a)}(\tilde{B}\tilde{Q}(s,a) - \tilde{Q}(s,a)) > \epsilon_a$ **or** $max_{(s,a)}(\tilde{Q}(s,a) - \tilde{B}\tilde{Q}(s,a)) > \epsilon_a$ **do**
14:             Set $\tilde{Q}(s,a) = \tilde{B}\tilde{Q}(s,a) \ \forall \ (s,a)$.
15:         **end while**
16:     **end if**
17: **end loop**
18: **function** $\tilde{B}\tilde{Q}(s,a)$
19:     **if** $|u(s,a)| \geq k_m$ **then**
20:         Let $(s,a,r_i,s_i')$ be the $i$-th sample in $u(s,a)$.
21:         **for** $j = 1$ **to** $k_m$ **do**
22:             $g(j) = \sum_{i=1+(j-1)\frac{|u(s,a)|}{k_m}}^{j\frac{|u(s,a)|}{k_m}}\left(r_i + \gamma\max_{\bar{a}}\tilde{Q}(s_i',\bar{a})\right)$
23:         **end for**
24:         **return** $\min\left\{Q_{\max}, \frac{\epsilon_b}{\sqrt{|u(s,a)|}} + \frac{k_m \, median\{g(1),...g(k_m)\}}{|u(s,a)|}\right\}$
25:     **else**
26:         **return** $Q_{\max}$.
27:     **end if**
28: **end function**

---

Algorithm 1 has three parameters that can be set by the user:

- $k$ is the maximum number of samples per state-action. As we will show, higher values for $k$ lead to increased sample complexity but better approximation.
- $\epsilon_a$ is an "acceptable error" term. Since Median-PAC is based on value iteration (lines 13 through 15) we specify a threshold after which value iteration should terminate. Value

iteration is suspended when the max-norm of the difference between Bellman backups is no larger than $\epsilon_a$.

- Due to the stochasticity of Markov decision processes, Median-PAC is only guaranteed to achieve a particular approximation quality with some probability. $k_m$ offers a trade-off between approximation quality and the probability that this approximation quality is achieved. For a fixed $k$ smaller values of $k_m$ offer potentially improved approximation quality, while larger values offer a higher probability of success. For simplicity of exposition our analysis requires that $k = 2^i k_m$ for some integer $i$. If $k_m \geq \left\lceil \frac{50}{9} \ln \frac{4 \log_2 \frac{4Q_{\max}^2}{\epsilon_a^2} |SA|^2}{\delta} \right\rceil$ the probability of failure is bounded above by $\delta$.

Like most modern PAC exploration algorithms, Median-PAC is based on the principle of optimism in the face of uncertainty. At every step, the algorithm selects an action greedily based on the current estimate of the $Q$-value function $\tilde{Q}$. The value function is optimistically initialized to $Q_{\max}$, the highest value that any state-action can take. If $k$ is set appropriately (see theorem 5.4), the value function is guaranteed to remain approximately optimistic (approximately represent the most optimistic world consistent with the algorithm's observations) with high probability.

We would like to draw the reader's attention to two aspects of Median-PAC, both in the way Bellman backups are computed: 1) Instead of taking a simple average over sample values, Median-PAC divides them into $k_m$ sets, computes the mean over each set, and takes the median of means. 2) Instead of using all the samples available for every state-action, Median-PAC uses samples in batches of a power of 2 times $k_m$ (line 9). The reasoning behind the first choice follows from the discussion above: using the median of means will allow us to show that Median-PAC's complexity scales with the variance of the Bellman operator (see definition 5.1) rather than $Q_{\max}^2$. The reasoning behind using samples in batches of increasing powers of 2 is more subtle. A key requirement in the analysis of our algorithm is that samples belonging to the same state-action are independent. While the outcome of sample $i$ does not provide information about the outcome of sample $j$ if $i < j$ (from the Markov property), the fact that $j$ samples exist can reveal information about the outcome of $i$. If the first $i$ samples led to a severe underestimation of the value of the state-action in question, it is likely that $j$ samples would never have been collected. The fact that they did gives us some information about the outcome of the first $i$ samples. Using samples in batches, and discarding the old batch when a new batch becomes available, ensures that the outcomes of samples within each batch are independent from one another.

## 5 Analysis

**Definition 5.1.** $\sigma$ *is the minimal constant satisfying*

$$\forall (s, a, \pi^{\tilde{Q}}, \tilde{Q}), \sqrt{\sum_{s'} p(s'|s, a) \Big( R(s, a, s') + \gamma \tilde{Q}(s', \pi^{\tilde{Q}}(s')) - B^{\pi^{\tilde{Q}}} \tilde{Q}(s, a) \Big)^2} \leq \sigma,$$

*where $\forall \tilde{Q}$ refers to any value function produced by Median-PAC, rather than any conceivable value function (similarly $\pi^{\tilde{Q}}$ refers to any greedy policy over $\tilde{Q}$ followed during the execution of Median-PAC rather than any conceivable policy).*

In the following we will call $\sigma^2$ the variance of the Bellman operator. Note that the variance of the Bellman operator is not the same as the variance, or stochasticity in the transition model of an MDP. A state-action can be highly stochastic (lead to many possible next states), yet if all the states it transitions to have similar values, the variance of its Bellman operator will be small.

From Lemmas 5.2, 5.3, and theorem 5.4 below, we have that Median-PAC is efficient PAC-MDP.

**Lemma 5.2.** *The space complexity of algorithm 1 is $O(k|S||A|)$.*

*Proof.* Follows directly from the fact that at most $k$ samples are stored per state-action. □

**Lemma 5.3.** *The per step computational complexity of algorithm 1 is bounded above by*

$$O\left( \frac{k|S||A|^2}{1 - \gamma} \ln \frac{Q_{\max}}{\epsilon_a} \right).$$

*Proof.* The proof of this lemma is deferred to the appendix. □

Theorem 5.4 below is the main theorem of this paper. It decomposes errors into the following three sources:

1. $\epsilon_a$ is the error caused by the fact that we are only finding an $\epsilon_a$-approximation, rather than the true fixed point of the approximate Bellman operator $\tilde{B}$, and the fact that we are using only a finite set of samples (at most $k$) to compute the median of the means, thus we only have an estimate.

2. $\epsilon_u$ is the error caused by underestimating the variance of the MDP. When $k$ is too small and Median-PAC fails to be optimistic, $\epsilon_u$ will be non-zero. $\epsilon_u$ is a measure of how far Median-PAC is from being optimistic (follow the greedy policy over the value function of the most optimistic world consistent with its observations).

3. Finally, $\epsilon_e(t)$ is the error caused by the fact that at time $t$ there may exist state-actions that do not yet have $k$ samples.

**Theorem 5.4.** *Let $(s_1, s_2, s_3, \ldots)$ be the random path generated on some execution of Median-PAC, and $\tilde{\pi}$ be the (non-stationary) policy followed by Median-PAC. Let $\epsilon_u = \max\{0, \sigma\sqrt{4k_m} - \epsilon_a\sqrt{k}\}$, and $\epsilon_a$ be defined as in algorithm 1. If $k_m = \left\lceil \frac{50}{9} \ln \frac{4\log_2 \frac{4Q_{\max}^2}{\epsilon_a^2}|SA|^2}{\delta} \right\rceil$,*

*$\epsilon_a \leq \frac{\epsilon_b}{\sqrt{k}}$, $\frac{2\left\lceil \frac{1}{1-\gamma} \ln \frac{(1-\gamma)Q_{\max}}{\epsilon_a} \right\rceil 2 \ln \frac{\log_2 \frac{2k}{k_m}}{\delta}}{k_m|SA|+1} < 1$, and $k = 2^i k_m$ for some integer $i$, then with probability at least $1 - \delta$, for all $t$*

$$V^*(s_t) - V^{\tilde{\pi}}(s_t) \leq \frac{2\epsilon_u + 5\epsilon_a}{1 - \gamma} + \epsilon_e(t), \tag{1}$$

*where*

$$\sum_{t=0}^{\infty} \epsilon_e(t) < c_0 \left( \left(2k_m + \log_2 \frac{2k}{k_m}\right) Q_{\max} + \epsilon_a k \left(8 + \frac{8}{\sqrt{2}}\right) \right), \tag{2}$$

*and*

$$c_0 = \frac{(|SA| + 1)\left(1 + \log_2\left\lceil \frac{1}{1-\gamma} \ln \frac{(1-\gamma)Q_{\max}}{\epsilon_a} \right\rceil\right)\left\lceil \frac{1}{1-\gamma} \ln \frac{(1-\gamma)Q_{\max}}{\epsilon_a} \right\rceil}{1 - \sqrt{\frac{2\left\lceil \frac{1}{1-\gamma} \ln \frac{(1-\gamma)Q_{\max}}{\epsilon_a} \right\rceil 2 \ln \frac{\log_2 \frac{2k}{k_m}}{\delta}}{k_m|SA|+1}}}.$$

*If $k = 2^i k_m$ where $i$ is the smallest integer such that $2^i \geq \frac{4\sigma^2}{\epsilon_a^2}$, and $\epsilon_0 = (1 - \gamma)\epsilon_a$, then with probability at least $1 - \delta$, for all $t$*

$$V^*(s_t) - V^{\tilde{\pi}}(s_t) \leq \epsilon_0 + \epsilon_e(t), \tag{3}$$

*where[5]*

$$\sum_{t=0}^{\infty} \epsilon_e(t) \approx \tilde{O}\left(\left(\frac{\sigma^2}{\epsilon_0(1-\gamma)^2} + \frac{Q_{\max}}{1-\gamma}\right)|SA|\right). \tag{4}$$

*Note that the probability of success holds for all timesteps simultaneously, and $\sum_{t=0}^{\infty} \epsilon_e(t)$ is an undiscounted infinite sum.*

*Proof.* The detailed proof of this theorem is deferred to the appendix. Here we provide a proof sketch:

The non-stationary policy of the algorithm can be broken up into fixed policy (and fixed approximate value function) segments. The first step in proving theorem 5.4 is to show that the Bellman error of each state-action at a particular fixed approximate value function segment is acceptable with respect to the number of samples currently available for that state-action with high probability. We use Cantelli's and McDiarmid's inequalities to prove this point. This is where the median of means

becomes useful, and the main difference between our work and earlier work. We then combine the result from the median of means, the fact that there are only a small number of possible policy and approximate value function changes that can happen during the lifetime of the algorithm, and the union bound, to prove that the Bellman error of all state-actions during all timesteps is acceptable with high probability. We subsequently prove that due to the optimistic nature of Median-PAC, at every time-step it will either perform well, or learn something new about the environment with high probability. Since there is only a finite number of things it can learn, the total cost of exploration for Median-PAC will be small with high probability. □

A typical "number of suboptimal steps" sample complexity bound follows as a simple corollary of theorem 5.4. If the total cost of exploration is $\sum_{t=0}^{\infty} \epsilon_e(t)$ for an $\epsilon_0$-optimal policy, there can be no more than $\frac{\sum_{t=0}^{\infty} \epsilon_e(t)}{\epsilon_1}$ steps that are more than $(\epsilon_0 + \epsilon_1)$-suboptimal.

Note that the sample complexity of Median-PAC depends log-linearly on $Q_{\max}$, which can be finite even if $R_{\max}$ is infinite. Consider for example an MDP for which the reward at every state-action follows a Gaussian distribution (for discrete MDPs this example requires rewards to be stochastic, while for continuous MDPs rewards can be a deterministic function of state-action-nextstate since there can be an infinite number of possible nextstates for every state-action). If the mean of the reward for every state-action is bounded above by $c$, $Q_{\max}$ is bounded above by $\frac{c}{1-\gamma}$, even though $R_{\max}$ is infinite.

As we can see from theorem 5.4, apart from being the first PAC exploration algorithm that can be applied to MDPs with unbounded rewards, Median-PAC offers significant advantages over the current state of the art for MDPs with bounded rewards. Until recently, the algorithm with the best known sample complexity for the discrete state-action setting was MORMAX, an algorithm by Szita and Szepesvári [16]. Theorem 5.4 offers an improvement of $\frac{1}{(1-\gamma)^2}$ even in the worst case, and trades a factor of $Q_{\max}^2$ for a (potentially much smaller) factor of $\sigma^2$. A recent algorithm by Pazis and Parr [12] currently offers the best known bounds for PAC exploration without additional assumptions on the number of states that each action can transition to. Compared to that work we trade a factor of $Q_{\max}^2$ for a factor of $\sigma^2$.

## 5.1 Using Median-PAC when $\sigma$ is not known

In many practical situations $\sigma$ will not be known. Instead the user will have a fixed exploration cost budget, a desired maximum probability of failure $\delta$, and a desired maximum error $\epsilon_a$. Given $\delta$ we can solve for the number of sets as $k_m = \left\lceil \frac{50}{9} \ln \frac{4 \log_2 \frac{4Q_{\max}^2 |SA|^2}{\epsilon_a^2}}{\delta} \right\rceil$, at which point all variables in equation 2 except for $k$ are known, and we can solve for $k$. When the sampling budget is large enough such that $k \geq \frac{4\sigma^2 k_m}{\epsilon_a^2}$, then $\epsilon_u$ in equation 1 will be zero. Otherwise $\epsilon_u = \sigma\sqrt{4k_m} - \epsilon_a\sqrt{k}$.

## 5.2 Beyond the discrete state-action setting

Recent work has extended PAC exploration to the continuous state [11] concurrent exploration [4] and delayed update [12] settings. The goal in the concurrent exploration setting is to explore in multiple identical or similar MDPs and incur low aggregate exploration cost over all MDPs. For a concurrent algorithm to offer an improvement over non-concurrent exploration, the aggregate cost must be lower than the cost of non-concurrent exploration times the number of tasks. The delayed update setting takes into account the fact that in real world domains, reaching a fixed point after collecting a new sample can take longer that the time between actions. Contrary to other work that has exploited the variance of MDPs to improve bounds on PAC exploration [7, 3] our analysis does not make assumptions about the number of possible next states from a given action. As such, Median-PAC and its bounds are easily extensible to the continuous state, concurrent exploration, delayed update setting. Replacing the average over samples in an approximation unit with the median of means over samples in an approximation unit in the algorithm of Pazis and Parr [12], improves their bounds (which are the best published bounds for PAC exploration in these settings) by $(R_{\max} + \gamma Q_{\max})^2$ while introducing a factor of $\sigma^2$.

# 6  Experimental evaluation

We compared Median-PAC against the algorithm of Pazis and Parr [12] on a simple 5 by 5 gridworld (see appendix for more details). The agent has four actions: move one square up, down, left, or right. All actions have a $1\%$ probability of self-transition with a reward of 100. Otherwise the agent moves in the chosen direction and receives a reward of 0, unless its action causes it to land on the top-right corner, in which case it receives a reward of 1. The world wraps around and the agent always starts at the center. The optimal policy for this domain is to take the shortest path to the top-right corner if at a state other than the top-right corner, and take any action while at the top-right corner.

While the probability of any individual sample being a self-transition is small, unless the number of samples per state-action is very large, the probability that there will exist at least one state-action with significantly more than $\frac{1}{100}$ sampled self-transitions is high. As a result, the naive average algorithm frequently produced a policy that maximized the probability of encountering state-actions with more than $\frac{1}{100}$ sampled self-transitions. By contrast, it is far less likely that there will exist a state-action for which at least half of the sets used by the median of means have more than $\frac{1}{100}$ sampled self-transitions. Median-PAC was able to consistently find the optimal policy.

# 7  Related Work

Maillard, Mann, and Mannor [8] present the distribution norm, a measure of hardness of an MDP. Similarly to our definition of the variance of the Bellman operator, the distribution norm does not directly depend on the stochasticity of the underlying transition model. It would be interesting to see if the distribution norm (or a similar concept) can be used to improve PAC exploration bounds for "easy" MDPs.

While to the best our knowledge our work is the first in PAC exploration for MDPs that introduces a measure of hardness for MDPs (the variance of the Bellman operator), measures of hardness have been previously used in regret analysis [6]. Such measures include the diameter of an MDP [6], the one way diameter [2], as well as the span [2]. These measures express how hard it is to reach any state of an MDP from any other state. A major advantage of sample complexity over regret is that finite diameter is not required to prove PAC bounds. Nevertheless, if introducing a requirement for a finite diameter could offer drastically improved PAC bounds, it may be worth the trade-off for certain classes of problems. Note that variance and diameter of an MDP appear to be orthogonal. One can construct examples of arbitrary diameter and then manipulate the variance by changing the reward function and/or discount factor.

Another measure of hardness which was recently introduced in regret analysis is the Eluder dimension. Osband and Van Roy [10] show that if an MDP can be parameterized within some known function class, regret bounds that scale with the dimensionality, rather than cardinality of the underlying MDP can be obtained. Like the diameter, the Eluder dimension appears to be orthogonal to the variance of the Bellman operator, potentially allowing for the two concepts to be combined.

Lattimore and Hutter [7] have presented an algorithm that can match the best known lower bounds for PAC exploration up to logarithmic factors for the case of discrete MDPs where every state-action can transition to at most two next states.

To the best of our knowledge there has been no work in learning with unbounded rewards. Harrison [5] has examined the feasibility of planning with unbounded rewards.

# Acknowledgments

We would like to thank Emma Brunskill, Tor Lattimore, and Christoph Dann for spotting an error in an earlier version of this paper, as well as the anonymous reviewers for helpful comments and suggestions. This material is based upon work supported in part by The Boeing Company, by ONR MURI Grant N000141110688, and by the National Science Foundation under Grant No. IIS-1218931. Opinions, findings, conclusions or recommendations herein are those of the authors and not necessarily those of the NSF.

## Footnotes

[1]Even though domains with truly unbounded rewards are not common, many domains exist for which infrequent events with extremely high (winning the lottery) or extremely low (nuclear power-plant meltdown) rewards exist. Algorithms whose sample complexity scales with the highest magnitude event are not well suited to such domains.

[2]For simplicity of exposition we assume that the same set of actions is available at every state. Our results readily extend to the case where the action set can differ from state to state.

[3]Note that $V^\pi(s_t)$ denotes the expected, discounted, accumulated reward of the arbitrarily complex policy $\pi$ from state $s_t$ at time $t$, rather than the expectation of some stationary snapshot of $\pi$.

[4]The number of samples per set was chosen so as to minimize the constants in the final bound.

[5] $f(n) = \tilde{O}(g(n))$ is a shorthand for $f(n) = O(g(n)\log^c g(n))$ for some constant $c$.

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
