[Supplementary Material]

# 8 Appendix A: Experimental evaluation

Figure 1: Accumulated discounted reward as a function of the number of episodes for a random walk, the algorithm of Pazis and Parr [12], and Median-PAC on a simple gridworld. Each plot represents an average over 1000 independent repetitions. Median-PAC significantly outperforms the algorithm of Pazis and Parr even for small values of $k_m$.

The discount factor for the gridworld described in section 6 was set to $0.98$, and every episode was 1000 steps long. We used modified versions of both learning algorithms that accumulate samples rather than using them in batches and discarding the old, smaller batch once a new batch has been collected. The algorithm of Pazis and Parr [12] (Average-PAC), was allowed allowed 1000 iterations of value iteration after each sample was added. Median-PAC was allowed 1000 iterations of value iteration every time the $i * k_m$-th sample was added to a state action, where $i > 0$ is an integer. $\epsilon_b$ was set to $0.01 Q_{\max}$ for both algorithms (since both algorithms truncate state-action values to $Q_{\max}$, setting $\epsilon_b$ close to $Q_{\max}$ for small values of $k$ saturates the value function). Notice that Median-PAC for $k = 105$ and $k_m = 21$ takes longer to achieve good performance than for $k = 9$ and $k_m = 3$. This is because for $k_m = 21$ the value of every state-action state is $Q_{\max}$ until at least 21 samples have been collected.

# 9 Appendix B: Analysis

Before we prove lemma 5.3 and theorem 5.4 we have to introduce a few supporting definitions and lemmas.

**Definition 9.1.** *Let* $|u(s,a)| = 2^i k_m$ *for some* $i \in \{1, 2, \dots\}$. *The function* $\boldsymbol{F^\pi(Q, u(s,a))}$ *is defined as*

$$F^\pi(Q, u(s,a)) = \frac{\epsilon_b}{\sqrt{|u(s,a)|}} + median\{G^\pi(Q, u(s,a), 1),$$

$$\dots,$$

$$G^\pi(Q, u(s,a), k_m)\},$$

*where*

$$G^\pi(Q, u(s,a), j) = \frac{k_m}{|u(s,a)|} \sum_{i=1+(j-1)\frac{|u(s,a)|}{k_m}}^{j\frac{|u(s,a)|}{k_m}} \left( r_i + \gamma Q(s_i', \pi(s_i')) \right),$$

*and* $(s, a, r_i, s_i')$ *is the i-th sample in* $u(s,a)$. *We will use* $\boldsymbol{F(Q, u(s,a))}$ *to denote* $F^{\pi^Q}(Q, u(s,a))$.

$F^\pi$ splits the samples in $u(s,a)$ into $k_m$ groups, computes the average of the sample values in each group, and returns the median of the averages.

**Definition 9.2.** *For state-action* $(s, a)$*, the approximate optimistic Bellman operator* $\boldsymbol{\tilde{B}^\pi}$ *for policy* $\pi$ *is defined as*

$$\tilde{B}^\pi Q(s,a) = \min\{Q_{\max}, F^\pi(Q, u(s,a))\}.$$

*We will use* $\boldsymbol{\tilde{B}Q(s,a)}$ *to denote* $\tilde{B}^{\pi^Q} Q(s,a)$. *When* $|u(s,a)| = 0$, $\tilde{B}^\pi Q(s,a) = Q_{\max}$.

The approximate optimistic Bellman operator is applied to the approximate value function on line 14 of the algorithm.

**Lemma 9.3.** $\tilde{B}$ *is a* $\gamma$-contraction in maximum norm.

*Proof.* Suppose $||Q_1 - Q_2||_\infty = \epsilon$. For any $(s, a)$ we have

$$\begin{aligned}
\tilde{B}Q_1(s,a) &= \min\{Q_{\max}, F(Q_1, u(s,a))\} \\
&\leq \min\{Q_{\max}, F(Q_2, u(s,a)) + \gamma\epsilon\} \\
&\leq \gamma\epsilon + \min\{Q_{\max}, F(Q_2, u(s,a))\} \\
&= \gamma\epsilon + \tilde{B}Q_2(s,a) \\
&\Rightarrow \tilde{B}Q_1(s,a) \leq \gamma\epsilon + \tilde{B}Q_2(s,a).
\end{aligned}$$

Similarly we have that $\tilde{B}Q_2(s,a) \leq \gamma\epsilon + \tilde{B}Q_1(s,a)$ which completes our proof. $\square$

**Lemma 9.4.** *Let* $\sigma$ *be defined as in Definition* 5.1. *For a fixed* $\tilde{Q}$ *and fixed* $(s,a)$ *such that* $|u(s,a)| > 0$

$$P\left(G^\pi(\tilde{Q}, u(s,a), j) - B^\pi \tilde{Q}(s,a) \leq -\frac{\sigma\sqrt{4k_m}}{\sqrt{|u(s,a)|}}\right) \leq \frac{1}{5},$$

*and*

$$P\left(G^\pi(\tilde{Q}, u(s,a), j) - B^\pi \tilde{Q}(s,a) \geq \frac{\sigma\sqrt{4k_m}}{\sqrt{|u(s,a)|}}\right) \leq \frac{1}{5}.$$

*Proof.* From Definition 9.1 we have that

$$B^\pi \tilde{Q}(s,a) = \mathrm{E}\left[G^\pi(\tilde{Q}, u(s,a), j)\right],$$

where the expectation is over the next-states that samples in $u(s,a)$ used by $G^\pi$ land on.

Let $Y$ be the set of $\frac{|u(s,a)|}{k_m}$ samples used by $G^\pi(\tilde{Q}, u(s,a), j)$ at $(s,a)$. Define $Z_1, \dots Z_{\frac{|u(s,a)|}{k_m}}$ to be random variables, one for each sample in $Y$. The distribution of $Z_i$ is the distribution of possible

values that $r_i + \gamma \max_{a'} \tilde{Q}(s'_i, a')$ can take. From the Markov property we have that $Z_1, \ldots Z_{\lfloor \frac{|u(s,a)|}{k_m} \rfloor}$ are independent random variables.[6] From Definition 5.1 we have that $var[Z_i] \leq \sigma^2 \ \forall \ i$, and $var[G^\pi(\tilde{Q}, u(s,a), j)] \leq \frac{\sigma^2 k_m}{|u(s,a)|}$.

From Cantelli's inequality we have

$$
P\left(G^\pi(\tilde{Q}, u(s,a), j) - B^\pi \tilde{Q}(s,a) \leq -\frac{\sigma\sqrt{4k_m}}{\sqrt{|u(s,a)|}}\right)
$$

$$
\leq P\left(G^\pi(\tilde{Q}, u(s,a), j) - \mathrm{E}\left[G^\pi(\tilde{Q}, u(s,a), j)\right] \leq -\frac{\sigma\sqrt{4k_m}}{\sqrt{|u(s,a)|}}\right)
$$

$$
\leq \frac{\frac{\sigma^2 k_m}{|u(s,a)|}}{\frac{\sigma^2 k_m}{|u(s,a)|} + \left(\frac{\sigma\sqrt{4k_m}}{\sqrt{|u(s,a)|}}\right)^2}
$$

$$
= \frac{\frac{\sigma^2 k_m}{|u(s,a)|}}{\frac{\sigma^2 k_m}{|u(s,a)|} + \frac{4\sigma^2 k_m}{|u(s,a)|}}
$$

$$
= \frac{1}{5},
$$

and

$$
P\left(G^\pi(\tilde{Q}, u(s,a), j) - B^\pi \tilde{Q}(s,a) \geq \frac{\sigma\sqrt{4k_m}}{\sqrt{|u(s,a)|}}\right)
$$

$$
\leq P\left(G^\pi(\tilde{Q}, u(s,a), j) - \mathrm{E}\left[G^\pi(\tilde{Q}, u(s,a), j)\right] \geq \frac{\sigma\sqrt{4k_m}}{\sqrt{|u(s,a)|}}\right)
$$

$$
\leq \frac{\frac{\sigma^2 k_m}{|u(s,a)|}}{\frac{\sigma^2 k_m}{|u(s,a)|} + \left(\frac{\sigma\sqrt{4k_m}}{\sqrt{|u(s,a)|}}\right)^2}
$$

$$
= \frac{\frac{\sigma^2 k_m}{|u(s,a)|}}{\frac{\sigma^2 k_m}{|u(s,a)|} + \frac{4\sigma^2 k_m}{|u(s,a)|}}
$$

$$
= \frac{1}{5}.
$$

$\square$

Based on Lemma 9.4 we can now bound the probability that an individual state-action will have Bellman error of unacceptably high magnitude for a particular $\tilde{Q}$:

**Lemma 9.5.** *Let $\sigma$ be defined as in Definition 5.1, and $\epsilon_u = \max\{0, \sigma\sqrt{4k_m} - \epsilon_b\}$. For a fixed $\tilde{Q}$*

$$
P\left(F^{\pi^*}(\tilde{Q}, u(s,a)) - B^{\pi^*}\tilde{Q}(s,a) \leq -\epsilon_u\right) \leq e^{-\frac{9k_m}{50}},
$$

*and*

$$
P\left(F^{\pi^{\bar{Q}}}(\tilde{Q}, u(s,a)) - B^{\pi^{\bar{Q}}}\tilde{Q}(s,a) \geq \epsilon_u + 2\frac{\epsilon_b}{\sqrt{|u(s,a)|}}\right) \leq e^{-\frac{9k_m}{50}}.
$$

*Proof.* Let $Y$ be the set of $|u(s,a)|$ samples used by $F^\pi(\tilde{Q}, u(s,a))$ at $(s,a)$. Define $Z_1, \ldots Z_{|u(s,a)|}$ to be random variables, one for each sample in $Y$. The distribution of $Z_i$ is the distribution of next

states $s_i'$, given $(s, a)$. From the Markov property, we have that $Z_1, \dots Z_{|u(s,a)|}$ are independent random variables (similarly to Lemma 9.4). Let $x_j$ be a realization of $X_j$, where $X_j$'s distribution is the joint distribution of all $Z_i$ corresponding to samples that participate in $G^\pi(\tilde{Q}, u(s, a), j)$.

We define $f^{\pi^*}(x_1, \dots x_{k_m})$ to be the function that counts the number of $j$'s such that

$$G^{\pi^*}(\tilde{Q}, u(s, a), j) - B^{\pi^*}\tilde{Q}(s, a) \leq -\frac{\sigma\sqrt{4k_m}}{\sqrt{|u(s, a)|}},$$

and $f^{\pi^{\tilde{Q}}}(x_1, \dots x_{k_m})$ to be the function that counts the number of $j$'s such that

$$G^{\pi^{\tilde{Q}}}(\tilde{Q}, u(s, a), j) - B^{\pi^{\tilde{Q}}}\tilde{Q}(s, a) \geq \frac{\sigma\sqrt{4k_m}}{\sqrt{|u(s, a)|}}.$$

From Lemma 9.4 we have that

$$E[f^{\pi^*}(x_1, \dots x_{k_m})] \leq \frac{k_m}{5},$$

and

$$E[f^{\pi^{\tilde{Q}}}(x_1, \dots x_{k_m})] \leq \frac{k_m}{5}.$$

$\forall\, i\, \in [1, k_m]$:

$$\sup_{x_1, \dots x_k, \hat{x}_i} |f^{\pi^*}(x_1, \dots x_{k_a}) - f^{\pi^*}(x_1, \dots, x_{i-1}\hat{x}_i, x_{i+1} \dots x_{|u(s,a)|})| \leq 1,$$

and

$$\sup_{x_1, \dots x_k, \hat{x}_i} |f^{\pi^{\tilde{Q}}}(x_1, \dots x_{k_a}) - f^{\pi^{\tilde{Q}}}(x_1, \dots, x_{i-1}\hat{x}_i, x_{i+1} \dots x_{|u(s,a)|})| \leq 1.$$

From McDiarmid's inequality we have

$$P\left(f^{\pi^*}(x_1, \dots x_{k_m}) \geq \frac{k_m}{2}\right) \leq P\left(f^{\pi^*}(x_1, \dots x_{k_m}) - E[f^{\pi^*}(x_1, \dots x_{k_m})] \geq \frac{3k_m}{10}\right)$$

$$\leq e^{-\frac{2\left(\frac{3k_m}{10}\right)^2}{k_m}}$$

$$= e^{-\frac{9k_m}{50}},$$

and

$$P\left(f^{\pi^{\tilde{Q}}}(x_1, \dots x_{k_m}) \geq \frac{k_m}{2}\right) \leq P\left(f^{\pi^{\tilde{Q}}}(x_1, \dots x_{k_m}) - E[f^{\pi^{\tilde{Q}}}(x_1, \dots x_{k_m})] \geq \frac{3k_m}{10}\right)$$

$$\leq e^{-\frac{2\left(\frac{3k_m}{10}\right)^2}{k_m}}$$

$$= e^{-\frac{9k_m}{50}}.$$

Since the probability that

$$G^{\pi^*}(\tilde{Q}, u(s, a), j) - B^{\pi^*}\tilde{Q}(s, a) \leq -\frac{\sigma\sqrt{4k_m}}{\sqrt{|u(s, a)|}}$$

for at least $\frac{k_m}{2}$ $j$'s is bounded above by $e^{-\frac{9k_m}{50}}$, and the probability that

$$G^{\pi^{\tilde{Q}}}(\tilde{Q}, u(s, a), j) - B^{\pi^{\tilde{Q}}}\tilde{Q}(s, a) \geq \frac{\sigma\sqrt{4k_m}}{\sqrt{|u(s, a)|}}$$

for at least $\frac{k_m}{2}$ $j$'s is bounded above by $e^{-\frac{9k_m}{50}}$, the result follows from Definition 9.1. $\square$

Given a bound on the probability that an individual state-action has Bellman error of unacceptably high magnitude, lemma 9.6 uses the union bound to bound the probability that there exists at least one state-action for some $\tilde{Q}$ produced by Median-PAC during execution, with Bellman error of unacceptably high magnitude.

**Lemma 9.6.** *Let $\epsilon_u = \max\{0, \sigma\sqrt{4k_m} - \epsilon_b\}$. The probability that for any $\tilde{Q}$ during an execution of Median-PAC there exists at least one $(s, a)$ with $|u(s, a)| > 0$ such that*

$$F^{\pi^*}(\tilde{Q}, u(s, a)) - B^{\pi^*}\tilde{Q}(s, a) \leq -\epsilon_u \tag{5}$$

*or*

$$F^{\pi^{\tilde{Q}}}(\tilde{Q}, u(s, a)) - B^{\pi^{\tilde{Q}}}\tilde{Q}(s, a) \geq \epsilon_u + 2\frac{\epsilon_b}{\sqrt{|u(s, a)|}} \tag{6}$$

*is bounded above by $2\log_2 \frac{4k}{k_m}|SA|^2 e^{-\frac{9k_m}{50}}$.*

*Proof.* At most $\log_2 \frac{4k}{k_m}|SA|$ distinct $\tilde{Q}$ exist for which $|u(s, a)| > 0$ for at least one $(s, a)$. Thus, there are at most $2\log_2 \frac{4k}{k_m}|SA|^2$ ways for at least one of the at most $|SA|$ state-actions to fail at least once during non-delay steps ($\log_2 \frac{4k}{k_m}|SA|^2$ ways each for equation 5 or equation 6 to be true at least once), each with a probability at most $e^{-\frac{9k_m}{50}}$. From the union bound, we have that the probability that for any $\tilde{Q}$ there exists at least one $(s, a)$ such that equation 5 or 6 is true, is bounded above by $2\log_2 \frac{4k}{k_m}|SA|^2 e^{-\frac{9k_m}{50}}$. $\square$

Based on Lemma 9.6 we can now bound the probability that any $(s, a)$ will have Bellman error of unacceptably high magnitude:

**Lemma 9.7.** *Let $\epsilon_u = \max\{0, \sigma\sqrt{4k_m} - \epsilon_b\}$. The probability that for any $\tilde{Q}$ during an execution of Median-PAC there exists at least one $(s, a)$ such that*

$$\tilde{Q}(s, a) - B^{\pi^*}\tilde{Q}(s, a) \leq -\epsilon_u - \epsilon_a \tag{7}$$

*or at least one $(s, a)$ with $|u(s, a)| > 0$ such that*

$$\tilde{Q}(s, a) - B^{\pi^{\tilde{Q}}}\tilde{Q}(s, a) \geq \epsilon_u + \epsilon_a + 2\frac{\epsilon_b}{\sqrt{|u(s, a)|}} \tag{8}$$

*is bounded above by $2\log_2 \frac{4k}{k_m}|SA|^2 e^{-\frac{9k_m}{50}}$.*

*Proof.* When $|u(s, a)| < k_m$, $\tilde{Q}(s, a) = Q_{\max}$. Since $B^{\pi^*}\tilde{Q}(s, a) \leq Q_{\max}$, $\tilde{Q}(s, a) - B^{\pi^*}\tilde{Q}(s, a) \leq -\epsilon_u - \epsilon_a$. Otherwise, $\forall(s, a, \tilde{Q})$ with probability $1 - 2\log_2 \frac{4k}{k_m}|SA|^2 e^{-\frac{9k_m}{50}}$

$$\begin{aligned}
B^{\pi^*}\tilde{Q}(s, a) &= \min\left\{Q_{\max}, B^{\pi^*}\tilde{Q}(s, a)\right\} \\
&< \min\left\{Q_{\max}, F^{\pi^*}(\tilde{Q}, u(s, a)) + \epsilon_u\right\} \\
&\leq \tilde{B}^{\pi^*}\tilde{Q}(s, a) + \epsilon_u \\
&\leq \tilde{B}\tilde{Q}(s, a) + \epsilon_u \\
&\leq \tilde{Q}(s, a) + \epsilon_u + \epsilon_a.
\end{aligned}$$

$\forall (s, a, \tilde{Q})$ with $u(s,a) \geq k_m$, with probability $1 - 2\log_2 \frac{4k}{k_m} |SA|^2 e^{-\frac{9k_m}{50}}$

$$
\begin{aligned}
B^{\pi^{\bar{Q}}} \tilde{Q}(s,a) &= \min \left\{ Q_{\max}, B^{\pi^{\bar{Q}}} \tilde{Q}(s,a) \right\} \\
&> \min \left\{ Q_{\max}, F^{\pi^{\bar{Q}}}(\tilde{Q}, u(s,a)) - \epsilon_u - 2\frac{\epsilon_b}{\sqrt{|u(s,a)|}} \right\} \\
&\geq \min \left\{ Q_{\max}, F^{\pi^{\bar{Q}}}(\tilde{Q}, u(s,a)) \right\} - \epsilon_u - 2\frac{\epsilon_b}{\sqrt{|u(s,a)|}} \\
&\geq \tilde{B}^{\pi^{\bar{Q}}} \tilde{Q}(s,a) - \epsilon_u - 2\frac{\epsilon_b}{\sqrt{|u(s,a)|}} \\
&= \tilde{B}\tilde{Q}(s,a) - \epsilon_u - 2\frac{\epsilon_b}{\sqrt{|u(s,a)|}} \\
&\geq \tilde{Q}(s,a) - \epsilon_u - \epsilon_a - 2\frac{\epsilon_b}{\sqrt{|u(s,a)|}}.
\end{aligned}
$$

Note that both the first half of Lemma 9.6 (used in the fist half of the proof) and the second half (used in the second half of the proof) hold simultaneously with probability $2\log_2 \frac{4k}{k_m} |SA|^2 e^{-\frac{9k_m}{50}}$, therefore we do not need to take a union bound over the individual probabilities. □

We will use the following three lemmas from Pazis and Parr (2016):

**Lemma 9.8.** *Let $t_i$ for $i = 0 \to l$ be the outcomes of independent (but not necessarily identically distributed) random variables in $\{0, 1\}$, with $P(t_i = 1) \geq p_i$. If $\frac{2}{m} \ln \frac{1}{\delta} < 1$ and*

$$
\sum_{i=0}^{l} p_i \geq \frac{m}{1 - \sqrt{\frac{2}{m} \ln \frac{1}{\delta}}},
$$

*then $\sum_{i=0}^{l} t_i \geq m$ with probability at least $1 - \delta$.*

**Lemma 9.9.** *Let $Q(s,a) - B^{\pi^*} Q(s,a) \geq -\epsilon_* \ \forall (s,a)$, $X_1, \ldots, X_i, \ldots, X_n$ be sets of state-actions where $Q(s,a) - B^{\pi^Q} Q(s,a) \leq \epsilon_i \ \forall (s,a) \in X_i$, $Q(s,a) - B^{\pi^Q} Q(s,a) \leq \epsilon_{\pi Q} \ \forall (s,a) \notin \cup_{i=1}^n X_i$, and $\epsilon_{\pi Q} \leq \epsilon_i \forall i$. Let $T_H = \left\lceil \frac{1}{1-\gamma} \ln \frac{(1-\gamma)Q_{\max}}{\epsilon_a} \right\rceil$ and define $H = \{1, 2, 4, \ldots, 2^i\}$ where $i$ is the largest integer such that $2^i \leq T_H$. Define $p_{h,i}(s)$ for $h \in [0, T_H-1]$ to be Bernoulli random variables expressing the probability of encountering exactly $h$ state-actions for which $(s,a) \in X_i$ when starting from state $s$ and following $\pi^Q$ for a total of $\min\{T, T_H\}$ steps. Finally let $p_{h,i}^e(s) = \sum_{m=h}^{2h-1} p_{m,i}(s)$. Then*

$$
V^*(s) - V^{\pi^Q}(s) \leq \frac{\epsilon_* + \epsilon_{\pi Q} + \epsilon_a}{1 - \gamma} + \epsilon_e,
$$

*where $\epsilon_e = 2\sum_{i=1}^{n} \left( \sum_{h \in H} \left( h p_{h,i}^e(s) \right) (\epsilon_i - \epsilon_{\pi Q}) \right) + \gamma^T Q_{\max}$.*

**Lemma 9.10.** *Let $\hat{B}$ be a $\gamma$-contraction with fixed point $\hat{Q}$, and $Q$ the output of*

$$
\frac{1}{1-\gamma} \ln \frac{Q_{\max}}{\epsilon}
$$

*iterations of value iteration using $\hat{B}$. Then if $0 \leq \hat{Q}(s,a) \leq Q_{\max}$ and $0 \leq Q_0(s,a) \leq Q_{\max} \ \forall (s,a)$, where $Q_0(s,a)$ is the initial value for $(s,a)$*

$$
-\epsilon \leq Q(s,a) - \hat{B}Q(s,a) \leq \epsilon \ \forall (s,a).
$$

Lemma 9.11 bounds the number of times the policy produced by Median-PAC can encounter state-actions with fewer than $k$ samples.

**Lemma 9.11.** *Let $(s_1, s_2, s_3, \ldots)$ be the random path generated on some execution of Algorithm 1. Let $\tau(t)$ be the number of steps from step $t$ to the next step for which the policy changes. Let $T_H = \left\lceil \frac{1}{1-\gamma} \ln \frac{(1-\gamma)Q_{\max}}{\epsilon_a} \right\rceil$ and define $H = \{1, 2, 4, \ldots, 2^i\}$ where $i$ is the largest such*

that $2^i \leq T_H$. Let $K_a = \{2^0 k_m, 2^1 k_m, 2^2 k_m, \ldots k\}$. *Let $k_a^-$ be the largest value in $K_a$ that is strictly smaller than $k_a$, or 0 if such a value does not exist. Let $X_{k_a}(t)$ be the set of state-actions at step $t$ for which $k_a^- = |u(s,a)|$. Define $p_{h,k_a}(s_t)$ for $k_a \in K_a$ to be Bernoulli random variables that express the following conditional probability: Given $\tilde{Q}$ at step $t$, exactly $h$ state-actions in $X_{k_a}(t)$ are encountered during the next $\min\{T_H, \tau(t)\}$ steps. Let $p_{h,k_a}^e(s_t) = \sum_{i=h}^{2h-1} p_{i,k_a}(s_t)$. If*

$$\frac{2\left\lceil \frac{1}{1-\gamma} \ln \frac{(1-\gamma)Q_{\max}}{\epsilon_a} \right\rceil 2 \ln \frac{\log_2 \frac{2k}{k_m}}{\delta}}{k_m|SA|+1} < 1, \text{ with probability at least } 1 - \frac{\delta}{2}$$

$$\sum_{t=0}^{\infty} \sum_{h \in H} (hp_{h,k_a}^e(s_{t,j})) < \frac{(k_a|SA|+1)\left(1 + \log_2\left\lceil \frac{1}{1-\gamma} \ln \frac{(1-\gamma)Q_{\max}}{\epsilon_a} \right\rceil\right)\left\lceil \frac{1}{1-\gamma} \ln \frac{(1-\gamma)Q_{\max}}{\epsilon_a} \right\rceil}{1 - \sqrt{\frac{2\left\lceil \frac{1}{1-\gamma} \ln \frac{(1-\gamma)Q_{\max}}{\epsilon_a} \right\rceil 2 \ln \frac{\log_2 \frac{2k}{k_m}}{\delta}}{k_m|SA|+1}}}$$

$\forall\, k_a \in K_a$ *and* $\forall\, h \in H$ *simultaneously.*

*Proof.* From the Markov property we have that $p_{h,k_a}^e(s_t)$ variables at least $T_H$ steps apart are independent.[7] Define $T_i^H$ for $i \in \{0, 1, \ldots, T_H - 1\}$ to be the (infinite) set of timesteps for which $t \in \{i, i+T_H, i+2T_H, \ldots\}$.

Since $k_a$ samples will be added to a state-action such that $|u(s,a)| = k_a^-$ before $|u(s,a)| = k_a$, at most $k_a|SA|$ state-actions such that $k_a^- = |u(s,a)|$ can be encountered.

Let us assume that there exists an $i \in \{0, 1, \ldots, T_H - 1\}$ and $h \in H$ such that

$$\sum_{t \in T_i^H} p_{h,k_a}^e(s_{t,j}) \geq \frac{k_a|SA|+1}{h\left(1 - \sqrt{\frac{2h}{k_a|SA|+1} \ln \frac{2\log_2 \frac{2k}{k_m}}{\delta}}\right)}.$$

From Lemma 9.8 it follows that with probability at least $1 - \frac{\delta}{2\log_2 \frac{2k}{k_m}}$, at least $k_a|SA|+1$ state-actions such that $k_a^- = |u(s,a)|$ will be encountered, which is a contradiction. It must therefore be the case that

$$\sum_{t \in T_i^H} p_{h,k_a}^e(s_{t,j}) < \frac{k_a|SA|+1}{h\left(1 - \sqrt{\frac{2h}{k_a|SA|+1} \ln \frac{2\log_2 \frac{2k}{k_m}}{\delta}}\right)}$$

with probability at least $1 - \frac{\delta}{2\log_2 \frac{2k}{k_m}}$ for all $i \in \{0, 1, \ldots, T_H - 1\}$ and $h \in H - \{T_H\}$ simultaneously, which implies that

$$\sum_{t=0}^{\infty} \sum_{h \in H} (hp_{h,k_a}^e(s_{t,j})) < \frac{(k_a|SA|+1)|H|T_H}{1 - \sqrt{\frac{2T_H}{k_a|SA|+1} \ln \frac{2\log_2 \frac{2k}{k_m}}{\delta}}}$$

$$\leq \frac{(k_a|SA|+1)\left(1 + \log_2\left\lceil \frac{1}{1-\gamma} \ln \frac{(1-\gamma)Q_{\max}}{\epsilon_a} \right\rceil\right)\left\lceil \frac{1}{1-\gamma} \ln \frac{(1-\gamma)Q_{\max}}{\epsilon_a} \right\rceil}{1 - \sqrt{\frac{2\left\lceil \frac{1}{1-\gamma} \ln \frac{(1-\gamma)Q_{\max}}{\epsilon_a} \right\rceil 2 \ln \frac{\log_2 \frac{2k}{k_m}}{\delta}}{k_m|SA|+1}}}$$

with probability at least $1 - \frac{\delta}{2\log_2 \frac{2k}{k_m}}$ for all $h \in H$ simultaneously.

From the union bound we have that since $k_a$ can take at most $\log_2 \frac{2k}{k_m}$ values, with probability $1 - \frac{\delta}{2}$

$$\sum_{t=0}^{\infty} \sum_{h \in H} (hp_{h,k_a}^e(s_{t,j})) < \frac{(k_a|SA|+1)\left(1 + \log_2\left\lceil \frac{1}{1-\gamma} \ln \frac{(1-\gamma)Q_{\max}}{\epsilon_a} \right\rceil\right)\left\lceil \frac{1}{1-\gamma} \ln \frac{(1-\gamma)Q_{\max}}{\epsilon_a} \right\rceil}{1 - \sqrt{\frac{2\left\lceil \frac{1}{1-\gamma} \ln \frac{(1-\gamma)Q_{\max}}{\epsilon_a} \right\rceil 2 \ln \frac{\log_2 \frac{2k}{k_m}}{\delta}}{k_m|SA|+1}}}$$

$\forall\, k_a \in K_a$ and $\forall\, h \in H$ simultaneously. $\qquad\square$

**Lemma 5.3.** *The per step computational complexity of algorithm 1 is bounded above by:*

$$O\left(\frac{k|S||A|^2}{1-\gamma}\ln\frac{Q_{\max}}{\epsilon_a}\right).$$

*Proof.* From lemma 9.10 we have that on every iteration of algorithm 1, lines 13 through 15 will we executed at most $O\left(\frac{1}{1-\gamma}\ln\frac{Q_{\max}}{\epsilon_a}\right)$ times. For each one of these iterations, function $\tilde{B}\tilde{Q}(s,a)$ will be called $|S||A|$ times. Line 22 in function $\tilde{B}\tilde{Q}(s,a)$ will be executed at most $k_m$ times, with a per execution cost of $O\left(\frac{k}{k_m}|A|\right)$. $\square$

**Theorem 5.4.** *Let $(s_1, s_2, s_3, \dots)$ be the random path generated on some execution of Median-PAC, and $\tilde{\pi}$ be the (non-stationary) policy followed by Median-PAC. Let $\epsilon_u = \max\{0, \sigma\sqrt{4k_m} - \epsilon_a\sqrt{k}\}$, and $\epsilon_a$ be defined as in algorithm 1. If $k_m = \left\lceil \frac{50}{9}\ln\frac{4\log_2\frac{4Q_{\max}^2}{\epsilon_a^2}|SA|^2}{\delta}\right\rceil$,*

$\epsilon_a \leq \frac{\epsilon_b}{\sqrt{k}}$, $\frac{2\left\lceil\frac{1}{1-\gamma}\ln\frac{(1-\gamma)Q_{\max}}{\epsilon_a}\right\rceil 2\ln\frac{\log_2\frac{2k}{k_m}}{\delta}}{k_m|SA|+1} < 1$, *and $k = 2^i k_m$ for some integer $i$, then with probability at least $1-\delta$, for all $t$*

$$V^*(s_t) - V^{\tilde{\pi}}(s_t) \leq \frac{2\epsilon_u + 5\epsilon_a}{1-\gamma} + \epsilon_e(t),$$

*where*

$$\sum_{t=0}^{\infty}\epsilon_e(t) < c_0\left(\left(2k_m + \log_2\frac{2k}{k_m}\right)Q_{\max} + \epsilon_a k\left(8 + \frac{8}{\sqrt{2}}\right)\right),$$

*and*

$$c_0 = \frac{(|SA|+1)\left(1 + \log_2\left\lceil\frac{1}{1-\gamma}\ln\frac{(1-\gamma)Q_{\max}}{\epsilon_a}\right\rceil\right)\left\lceil\frac{1}{1-\gamma}\ln\frac{(1-\gamma)Q_{\max}}{\epsilon_a}\right\rceil}{1 - \sqrt{\frac{2\left\lceil\frac{1}{1-\gamma}\ln\frac{(1-\gamma)Q_{\max}}{\epsilon_a}\right\rceil 2\ln\frac{\log_2\frac{2k}{k_m}}{\delta}}{k_m|SA|+1}}}.$$

*If $k = 2^i k_m$ where $i$ is the smallest integer such that $2^i \geq \frac{4\sigma^2}{\epsilon_a^2}$, and $\epsilon_0 = (1-\gamma)\epsilon_a$, then with probability at least $1-\delta$, for all $t$*

$$V^*(s_t) - V^{\tilde{\pi}}(s_t) \leq \epsilon_0 + \epsilon_e(t),$$

*where[8]*

$$\sum_{t=0}^{\infty}\epsilon_e(t) \approx \tilde{O}\left(\left(\frac{\sigma^2}{\epsilon_0(1-\gamma)^2} + \frac{Q_{\max}}{1-\gamma}\right)|SA|\right).$$

*Note that the probability of success holds for all timesteps simultaneously, and $\sum_{t=0}^{\infty}\epsilon_e(t)$ is an undiscounted infinite sum.*

*Proof.* From Lemma 9.7 we have that with probability at least $1 - 2\log_2\frac{4k}{k_m}|SA|^2 e^{-\frac{9k_m}{50}}$

$$\tilde{Q}(s,a) - B^{\pi^*}\tilde{Q}(s,a) > -\epsilon_u - \epsilon_a \tag{9}$$

for all $(s, a, \tilde{Q})$, and

$$\tilde{Q}(s,a) - B^{\pi^{\tilde{Q}}}\tilde{Q}(s,a) < \epsilon_u + \epsilon_a + 2\frac{\epsilon_b}{\sqrt{|u(s,a)|}} \tag{10}$$

for all $(s, a)$ with $|u(s,a)| \geq k_m$. We also have that

$$\tilde{Q}(s,a) - B^{\pi^{\tilde{Q}}}\tilde{Q}(s,a) \leq Q_{\max}\ \forall\ (s,a,\tilde{Q}).$$

Let $K_a$, $k_a^-$, $T_H$, $H$, $\tau(t)$, and $p_{h,k_a}^e(s_t)$ be defined as in lemma 9.11. With probability at least $1 - 2\log_2 \frac{4k}{k_m}|SA|^2 e^{-\frac{9k_m}{50}}$, for any $(s,a)$ with $|u(s,a)| > 0$ samples

$$\tilde{Q}(s,a) - B^{\pi^{\tilde{Q}}}\tilde{Q}(s,a) < \epsilon_u + \epsilon_a + 2\frac{\epsilon_b}{\sqrt{|u(s,a)|}}.$$

Even though $\tilde{\pi}$ is non-stationary, it is comprised of stationary segments. Starting from step $t$, $\tilde{\pi}$ is stationary for at least $\tau(t)$ steps. Substituting the above into Lemma 9.9 we have that with probability at least $1 - 2\log_2 \frac{4k}{k_m}|SA|^2 e^{-\frac{9k_m}{50}}$

$$V^*(s_t) - V^{\tilde{\pi}}(s_t) \leq \frac{2\epsilon_u + 3\epsilon_a + 2\frac{\epsilon_b}{\sqrt{k}}}{1-\gamma} + \epsilon_e(t),$$

where

$$\epsilon_e(t) = \gamma^{\tau(t)}Q_{\max} + 2\sum_{h\in H}(hp_{h,k_m}^e(s_t))Q_{\max} + \sum_{k_a\in\{K_a-k_m\}}2\sum_{h\in H}(hp_{h,k_a}^e(s_t))2\frac{\epsilon_b}{\sqrt{k_a}}.$$

From the above it follows that

$$\sum_{t=0}^{\infty}\epsilon_e(t)$$

$$= \sum_{t=0}^{\infty}\left(\gamma^{\tau(t)}Q_{\max} + 2\sum_{h\in H}(hp_{h,1}^e(s_{t,j}))Q_{\max} + \sum_{k_a\in\{K_a-k_m\}}2\sum_{h\in H}(hp_{h,k_a}^e(s_{t,j}))2\frac{\epsilon_b}{\sqrt{k_a}}\right)$$

$$= \sum_{t=0}^{\infty}\gamma^{\tau(t)}Q_{\max} + 2\sum_{t=0}^{\infty}\sum_{h\in H}(hp_{h,1}^e(s_{t,j}))Q_{\max} + 2\sum_{k_a\in\{K_a-k_m\}}\sum_{t=0}^{\infty}\sum_{h\in H}(hp_{h,k_a}^e(s_{t,j}))2\frac{\epsilon_b}{\sqrt{k_a}}$$

$$< \frac{|SA|Q_{\max}\log_2\frac{2k}{k_m}}{(1-\gamma)} + 2k_m c_0 Q_{\max} + 2\sum_{k_a\in\{K_a-k_m\}}k_a c_0 2\frac{\epsilon_b}{\sqrt{k_a}}$$

$$< \left(2k_m + \log_2\frac{2k}{k_m}\right)c_0 Q_{\max} + 2\sum_{k_a\in\{K_a-k_m\}}k_a c_0 2\frac{\epsilon_b}{\sqrt{k_a}}$$

$$= \left(2k_m + \log_2\frac{2k}{k_m}\right)c_0 Q_{\max} + 4c_0\epsilon_b\sum_{k_a\in\{K_a-k_m\}}\sqrt{k_a}$$

$$< \left(2k_m + \log_2\frac{2k}{k_m}\right)c_0 Q_{\max} + 4c_0\epsilon_b\sqrt{k}\left(\sum_{i=0}^{\infty}\left(\frac{1}{2^i} + \frac{1}{2^i\sqrt{2}}\right)\right)$$

$$= c_0\left(\left(2k_m + \log_2\frac{2k}{k_m}\right)Q_{\max} + \epsilon_b\sqrt{k}\left(8 + \frac{8}{\sqrt{2}}\right)\right)$$

with probability $1 - \delta - 2\log_2\frac{4k}{k_m}|SA|^2 e^{-\frac{9k_m}{50}}$, where in step 3 we used the fact that there can be at most $\log_2\frac{2k}{k_m}|SA|$ policy changes. Since Lemma 9.7 (used to bound the Bellman error of each $(s,a,\tilde{Q})$) holds with probability at least $1 - 2\log_2\frac{4k}{k_m}|SA|^2 e^{-\frac{9k_m}{50}}$ and Lemma 9.11 (used to bound how many times each $(s,a,\tilde{Q})$ is encountered) holds with probability of at least $1 - \frac{\delta}{2}$, the bound above holds with probability of at least $1 - \frac{\delta}{2} - 2\log_2\frac{4k}{k_m}|SA|^2 e^{-\frac{9k_m}{50}}$.

Setting $\epsilon_b = \epsilon_a\sqrt{k}$ we have that with probability at least $1 - \delta$

$$V^*(s_t) - V^{\tilde{\pi}}(s_t) \leq \frac{2\epsilon_u + 5\epsilon_a}{1-\gamma} + \epsilon_e(t),$$

where

$$\sum_{t=0}^{\infty}\epsilon_e(t) < c_0\left(\left(2k_m + \log_2\frac{2k}{k_m}\right)Q_{\max} + \epsilon_a k\left(8 + \frac{8}{\sqrt{2}}\right)\right).$$

Equations 3 and 4 follow by substitution and by using the fact that $\sigma \leq \frac{Q_{\max}}{2}$. $\qquad\square$

## Footnotes

[6]The state-actions the samples originate from as well as $\tilde{Q}$ and the transition model of the MDP are fixed with respect to $Z_i$, and no assumptions are made about their distribution. The only source of randomness is the the transition model of the MDP.

[7]While what happens at step $t$ affects which variables are selected at future timesteps, this is not a problem. We only care that the *outcomes* of the variables are independent given their selection.

[8] $f(n) = \tilde{O}(g(n))$ is a shorthand for $f(n) = O(g(n)\log^c g(n))$ for some $c$.