[Reviews · NeurIPS 2016]

Reviewer 1

Summary

The authors present an PAC RL algorithm that uses a median-of-means estimator to do bellman backups. They prove that this gives reduced dependence of performance on the range of Q-values, while introducing dependence the variability in next-state Q-values.

Qualitative Assessment

This is an excellent paper. It's beautifully written, I learned a new trick, and the results will be useful to many in this area going forward. The obvious missing piece in the paper is a connection to a domain that illustrates the properties that the authors claim are true about "domains of practical interest." They claim: "...the variance of the Bellman operator is rather small in many domains of practical interest. For example, this is true in many control tasks: It is not very often that an action takes us to the best possible state with 50% probability and to the worst possible state with 50% probability." Show me. The paper is under 7 pages long. There is room even just to look at a toy example and make the argument more compelling. Further to that point, what domains would be problematic? Cliff walker? What others? I am confident that these authors can present meaningful empirical results that really support the conclusions of the paper and teach the reader about the significance of their work -- beyond the usual "our line is higher than their line." Minor: Comment on the time complexity of Algorithm 1? 156: B^\pi \tilde Q (s,a) is a (deterministic) scalar. I think I know what you mean by the "variance" of that quantity -- it's the variance of the terms in the sum that define B^\pi \tilde Q (s,a) as per line 58. Is there any way to make the notation at Definition 5.1 less abusive?

Confidence in this Review

2-Confident (read it all; understood it all reasonably well)


Reviewer 2

Summary

This paper presents a new approach to PAC-MDP exploration in reinforcement learning. Their new algorithm Median-PAC uses a median of means estimator to replace a worst-case Q_max^2 with a problem specific sigma^2 = variance of the Q-value function. This result is an important step forward in the analysis of efficient exploration. ------------------- I have read the authors' rebuttal; I do not change my view on the paper significantly.

Qualitative Assessment

I like this paper and think it deserves to be accepted. - This paper addresses an important question for PAC-optimal exploration: problem specific bounds which go beyond the "worst-case". Replacing Q_max^2 with sigma^2 is an important breakthrough in the quality of PAC-bounds. - The application of median of means appears to be novel to efficient exploration. It may provide inspiration and robustness to similar settings beyond this individual paper. - The paper is relatively clear and easy to follow. That said, there are several elements which I believe could do with some improvement. - Overall, the discussion of the related work is not great. This is particularly disappointing since the paper is well under page limit, so it should have been possible. However, I do believe that it will be relatively easy for the authors to remedy this before the final camera ready version. - A similar notion to the variance sigma^2 is given by the span of an MDP (Tewari+Bartlett, 2009). Their resulting algorithm REGAL is not computationally tractable, however it should certainly be discussed in the context of this paper. There are other practical algorithms, such as PSRL (Osband+VanRoy, 2014a;b) which present Bayesian regret bounds in terms of the *span* or *smoothness* of the unknown optimal problem. The spirit of these bounds appears to be similar to this variance notion in the PAC setting. A proper discussion of these algorithms is absolutely crucial. - This paper does not include any experiments or simulations. This isn't a deal-breaker for me, but it would be a nice tool to help build intuition and demonstrate the potential applicability of this result. Perhaps some simple example could be designed to highlight the potential improvement of this algorithm. I think this would significantly add to the impact of this work for many people. - Formatting: I don't think that equations/variables should be bolded; terms such as "max" within formulas should be written {\rm max} or similar; the bellman operator is usually depicted by some kind of T, in this context perhaps \mathcal{T}^\pi would be better. Overall I like this paper and I would argue for its acceptance. I do think that a better treatment of related work is necessary and some computational experiments would be desirable.

Confidence in this Review

2-Confident (read it all; understood it all reasonably well)


Reviewer 3

Summary

The paper describes how to use the median of means method in estimating the Q and value functions of an MDP. The obtain stronger convergence guarantees even in MDPs with unbounded rewards.

Qualitative Assessment

To the best of my knowledge, the paper is the first to a) use the median of means algorithm in RL/MDPs b) derive bounds for MDPs with unbounded rewards c) not have a dependence in M (the max reward) in the bounds. While the median of means algorithm is well known, it has only recently been exploited as much as it should be. I am somewhat concerned that there may be an outburst of similar papers for various sgd type algorithms. The paper is well written and understandable. Line 87-88 : Should X be X' Algorithm 1 , Line 18 : s_i, a_i should be s,a ? Line 147-148 and Line 183-184 : The fact that the algorithm can be broken up into fixed policy segments seems very crucial to the proofs. It took me a while to figure out what in the algorithm assures this (step 16 in the algorithm). While this is a standard growing window trick, it is well hidden. It may be a good idea to elaborate this important point. Line 219-220: The extension to continuous state does seem non-trivial (though I am not yet familiar with reference [10]). Elaborating on that here, along with the improved results would add more heft to this paper.

Confidence in this Review

2-Confident (read it all; understood it all reasonably well)


Reviewer 4

Summary

This paper applies median of means technique to PAC-optimal exploration algorithm for MDPs. The authors proposed a new algorithm called Median-PAC and showed the sample complexity bound depending on the variance of Bellman operator.

Qualitative Assessment

This paper is interesting and well-written. Even it's a theoretical paper and the readers can follow the authors without reading the proofs. I like this written style and the way of telling story. But I have some questions and suggestions about this paper. 1) The variance of Bellman operator is a key metric in the paper. The authors mentioned that "Note that the variance of the Bellman operator is not the same as the variance, or stochasticity in the transition model of an MDP. A state-action can be highly stochastic (lead to many possible next states), yet if all the states it transitions to have similar values, the variance of its Bellman operator will be small." From the definition of 5.1, the variance is determined by structure of the MDPs. This is a pretty interesting metric of MDP. Do the authors have any idea about the connection or relation between other metrics (e.g. diameter or one-way diameter)? Furthermore, is there any practical way to determine the variance of Bellman operator, given a MDP? 2) The authors didn't provide any empirical results about their new algorithm. This slightly makes me disappointed. So, I lower the potential usefulness score. Although this is a theoretical paper, the authors mentioned about their motivation in applications. If the authors are able to provide some empirical results, it would be a good practical support for their new algorithm. In all, this paper is interesting and well-written. Readers without strong math background are still able to follow the authors. I suggest to accept.

Confidence in this Review

2-Confident (read it all; understood it all reasonably well)


Reviewer 5

Summary

The paper proposes a PAC algorithm for reinforcement learning in MDPs with potentially finite state and action spaces. The key difference to existing work is that this method uses median of means instead of the usual empirical mean to estimate the expected value of the next state in the Bellman operator. This allows to provide PAC bounds (in the form of TCE bounds) that only depend log-linearly on the range of the value function while introducing dependency on the variance of the values of successor states.

Qualitative Assessment

The authors' feedback clarified misunderstandings I had and addressed my concerns about usefulness of the presented results. With the remaining issues with clarity being removed, this paper can be a really nice contribution for the NIPS community. Original Review: ---------------- The idea of using the medians of means estimator to obtain stronger PAC guarantees for reinforcement learning algorithms seems very appealing to me and the presented results appear plausible. Yet, I see two main issues with this paper: First, there is a lack of clarity and insufficient explanations and second, the usefulness of the presented PAC guarantee is not entirely clear to me. Usefulness: * The authors state in line 40 on page 2 that their algorithm does not depend on the number of successor states which can even be unbounded. However, the sample complexity bound depends linearly on the number of states. For unbounded number of successor states and therefore unbounded number of states, the sample complexity bound becomes meaningless. * Definition 2.1 states that Q_max is an upper bound on the value of any state (for any policy I assume) but no assumptions have been made about a lower bound. I therefore wonder whether adding a constant to every reward (which should not affect the difficulty of the RL problem) affects the bound in Theorem 1 or not. Is it assumed that rewards / all values are nonnegative? * Definition 5.1 requires sigma^2 to be an upper bound on the variance of the value of the next state for any value-function and policy. This seems rather demanding and raises the question when sigma^2 is actually significantly smaller than (Q_max/2)^2 especially discount factors close to 1. For gamma close to 1, it is sufficient that there is a (s,a) pair and a subset S' of S so that p(s' in S' | s, a) \approx 1/2 for sigma to be close to (Q_max/2)^2 (by choosing \tilde Q and pi appropriately). This does seem to be the case in most relevant MDPs. Clarity and presentation: * The paper claims the algorithm is PAC-optimal (w.g. line 200). However, a direct comparison to a lower-bound is not provided which makes it not clear to me why this algorithm has indeed optimal (= matching a lower bound up to constants and log-terms) sample-complexity. * Knowing a nontrivial bound on sigma is unrealistic, especially ahead of executing the algorithm. However, the current bound assumes parameters k, k_m to be chosen depending on sigma. The bound in Theorem 5.2 is therefore rather artificial. It is stated in line 199 that the bound can also be formulated for the more relevant case of choosing k, k_m fixed but not actually provided in the paper or appendix. * The quantities k_m, k and epsilon_s are only defined implicitly (e.g. k_m appears on both sides of the equality sign. It is therefore hard to identify how changes in problem characteristics affect the provided sample complexity bound. These dependencies should be disentangled to make the dependency of the bound on the important quantities: 1/(1-gamma), |S|, |A|, Q_max and sigma clear. * The provided proof sketch discusses the general structure which is very similar to the analysis of most other Optimism-in-the-face-of-uncertainty methods. However, it provides no insight where the median of means actually changes the proof and allows (potentially) tighter bounds. While Section 3 is very helpful for understanding why median of means can be beneficial, more details on how this is leveraged in PAC RL would be very much appreciated. Minor things: * Should the value of of n in line 110.5 not depend on 2/delta instead of 1/delta as each of the two failure cases has to be bounded by 1- delta/2? * Algorithm 1: \epsilon_a is also an input / parameter of the algorithm * Line 20 of Algorithm 1 redefines i in the sum, which is already defined in line 16

Confidence in this Review

2-Confident (read it all; understood it all reasonably well)


Reviewer 6

Summary

The paper presents a PAC-optimal exploration algorithm with a performance bound that depends on the variance of the Bellman operator, rather than the range of the Q-values. This is done by applying the median of means method.

Qualitative Assessment

=== Review Summary === I like the paper, but it could use more motivation / elaboration of certain design choices and results. === Technical quality === The size of the sets being 4\sigma^2 / \epsilon^2 seemed completely arbitrary to me. Is there a reason for the choice of the constant to be 4? If so, it needs to be stated. One of the things I found confusing is the authors claiming that their algorithm can deal with unbounded rewards, yet having a Q_max term, both in the bound, and the algorithm. If sigma is small, wouldn’t Qmax actually dominate the bound? It was then a bit hard to keep track of all the parameters. k_m is taken to be an input to the algorithm, but how can one expect to know it in advance? In general there are a lot of arguments into each choice of variable (in Theorem 5.2 e.g.), and it is not always clear what is an input to fix, and what needs to be derived. There is discussion in Lines 198-199 that suggests that one can fix k and k_m, derive \epsilon_s as a function of sigma, and apply the method. More elaboration would be helpful. In the bullet points starting at Line 118, there are promises to demonstrate the impact of k, k_m, and epsilon_b. Lines 134-135 also contain a promise to specify the proper choices of epsilon_b in Theorem 5.2. These are never fulfilled later. Given the complex relationships of the bound with the parameters, and multiple substitutions, an explicit discussion after the theorem would be very beneficial. === Novelty/originality === As far as I know, this line of analysis in the context of RL is novel. === Potential impact or usefulness === The paper states the divide between the overly pessimistic PAC-optimal exploration theory and practice as a motivation for the method, and sets out to to make the first step to bridge it. To illustrate the practical relevance, an experiment illustrating the more faithful bound for an MDP with a low-variance Bellman operator, would have been great. === Clarity and presentation === The first half of the paper is very well-written and clear. I appreciated the clarity of Section 3 (though it could seem basic to someone better familiar with PAC conventions). The second half could use more clarity. === Minor comments / proofreading === Line 74: Footnote mark after punctuation Lines 83, 89: \epsilon-close (with dash) Section 3: Is there a reason to not use absolute value in the probability of error being within \epsilon? It kept distracting me. Line 90: RL acronym is never introduced Line 109: the base of the exponent should be e, rather than \epsilon? Algorithm 1, Line 1: \epsilon_a needs to be an input? Line 109: an extra clarifying sentence here would be good to remind the readers that because the median is considered the probabilities carry the same meaning as those of the deviation from the mean from before, and that k_m = n\epsilon^2 / (4\sigma^2) Line 156: It isn’t necessary to specify that \sigma is non-negative, the expression involving the square root implies it. Line 231: The phrasing in this sentence is weird, since the paper does not emphasize “trying to distinguish between easy and hard problems” elsewhere.

Confidence in this Review

2-Confident (read it all; understood it all reasonably well)